# Learning Treatment Allocations with Risk Control Under Partial Identifiability

**Sofia Ek** [1]   **Dave Zachariah** [1]

## Abstract

Learning beneficial treatment allocations for a patient population is an important problem in precision medicine. For such allocations, a certain proportion of treated patients may not receive any benefit. This proportion of unnecessary treated represents a 'treatment risk' which is a waste of resources and may, in addition, expose patients to unnecessary adverse effects. Therefore, we aim to control the treatment risk when learning beneficial allocations. This learning problem is complicated by the fact that the treatment risk is generally not identifiable from either randomized trial or observational data. We propose a certifiable learning method that controls treatment risk, using finite samples in the partially identified setting. The method is illustrated using both simulated and real data.

## 1. Introduction

The allocation of treatments in a patient population is a fundamental challenge in precision medicine. Consider a *policy* $\pi(X)$ that recommends one of two treatment options $A \in \{0, 1\}$ based on a patient's observable characteristics $X$. The recommendation may be to provide standard care ($A = 0$) or administer aggressive treatment ($A = 1$) among cancer patients. Here we consider dichotomous health outcomes, such as recovery or nonrecovery within a given period. Let the binary loss $L \in \{0, 1\}$ indicate if the outcome is non-beneficial. The proportion of such outcomes, $\mathbb{P}_\pi(L = 1)$, defines the *population risk* under a policy.

The population risk is minimized by any policy $\pi(X)$ that assigns $A = 1$ to patient covariates $X$ for which the probability of health loss is lower than under $A = 0$. However, this can easily result in a large proportion of treated *who receive no benefit*. For a simple illustration, consider a co-

variate $X \in [30, 80]$ to be the age of a patient in years. Suppose the (conditional) probabilities of health loss under the two treatment options are

$$p(L = 1|A = 1, X) = 0.01 \cdot (X - 30)$$
$$< p(L = 1|A = 0, X) = 0.80,$$

for all $X$. Here the treat-all policy $\pi(X) \equiv 1$ minimizes the population risk, but then a quarter of all 55-year olds and half of all 80-year-olds receive no benefit from their treatments. Interventions that do not benefit a patient are an increasing clinical concern (Brownlee et al., 2017; Lyu et al., 2017; Ooi, 2020) because they waste resources and can, additionally, expose patients to unnecessary adverse effects that may violate the principle of non-maleficence (i.e., "above all, do no harm") (Smith, 2005).

The proportion of treated *who receive no benefit*, $\mathbb{P}_\pi(L = 1|A = 1)$, we denote as the *treatment risk* of policy $\pi$. In this paper, we aim to learn policies that constrain the treatment risk to some specified tolerance $\tau$, while minimizing the overall population risk. That is, with high probability the treatment risk is no greater than $\tau$, as illustrated in Figure 1. Varying $\tau$ controls a trade-off between treatment and population risks; as the tolerance decreases the learned allocations $\pi(X)$ focuses on those patient characteristics $X$ that are most likely to yield benefits under treatment.

The challenge we consider in this paper is to learn a policy $\pi$ with certified risk control from finite data, even in circumstances where its risk is *not point identifiable*. That is, when the sampling process is not informative enough to determine a unique value of the risk (Manski, 2003; 2007). For *observational* data, this arises when there are unmeasured confounders, which in general cannot be assessed (Kallus, 2018). For *randomized trial* data, this occurs when the trial and intended populations differ (Westreich, 2019).

The main contribution of this work is a method for learning treatment allocation policies that aims at reducing the population risk,

- while *controlling* the treatment risk with a high probability in a *finite* sample setting, and that is

- *valid* even under *partial* identifiability in either observational or randomized trial data.

[1]Div. Systems and Control, Dept. Information Technology, Uppsala University, Sweden.. Correspondence to: Dave Zachariah <dave.zachariah@it.uu.se>.

*Proceedings of the 43rd International Conference on Machine Learning*, Seoul, South Korea. PMLR 306, 2026. Copyright 2026 by the author(s).

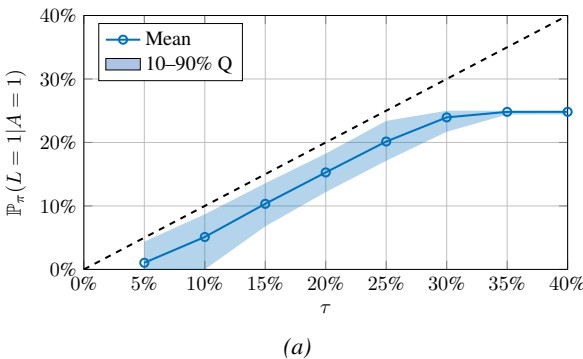

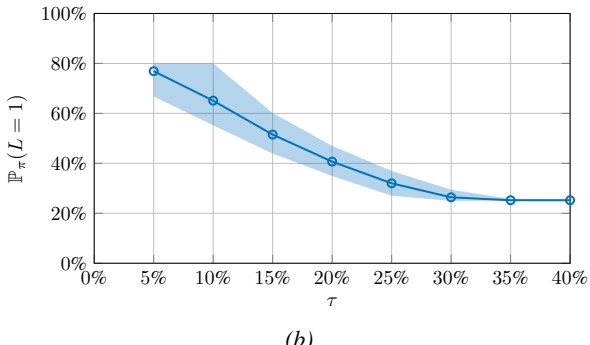

*(a)*

*(b)*

*Figure 1.* (a) Proportion of treated with no benefit versus specified tolerance $\tau$ using learned allocation policies $\pi$. (b) Overall proportion with non-beneficial outcomes versus $\tau$. The proposed learning method limits (a) to be no greater than $\tau$ with a probability of at least 90%, while minimizing (b). The shaded regions (10-90th percentiles) represent the resulting risks of policies learned from 1000 different datasets. Thus $\tau$ trades off two types of risks (a) and (b). The details of the example are given in Section 5.1.

The paper is structured as follows: Section 2 introduces the problem, followed by a discussion of its connection to existing literature in Section 3. We develop a policy learning method in Section 4, which is evaluated using both synthetic and real-world data in Section 5. Finally, Section 6 provides a discussion of the properties of the method.

## 2. Problem Formulation

A policy $\pi(X)$ allocates treatments $A$ based on individual covariates $X$ which may affect the binary outcome $L$. The application of the allocation policy $\pi$ to an intended patient population can be described using variables $(X, U, A, L)$ drawn from nonparametric structural casual model with a directed acyclic graph given in Figure 2a (Pearl, 2009; Peters et al., 2017). Here $U$ denotes additional *unobserved* factors which also affect $L$ and $S = 0$ indicates *sampling* from the patient population. ($S = 1$ will denote sampling from a trial population below, cf. Westreich et al. (2017).)

For notational convenience, we use formalism employed in the literature on off-policy learning for contextual bandits (Dudík et al., 2011; Swaminathan & Joachims, 2015). Specifically, the joint distribution of the decision process under $\pi$ can be *causally* factored as

$$p_\pi(X, U, A, L | S = 0) = \\ p(L|A, X, U) \, \mathbb{1}(A = \pi(X)) \, p(X, U | S = 0), \quad (1)$$

where $p(X, U | S = 0)$ describes the population characteristics, $\mathbb{1}(A = \pi(X))$ is the assignment by the policy and the causal effects of the treatments are captured by $p(L|A, X, U)$. From the joint distribution (1), we can express the *population risk* under $\pi$, i.e., the overall propor-

tion of health losses in the patient population ($S = 0$), as

$$R(\pi) \equiv \mathbb{P}_\pi(L = 1 | S = 0) \\ = \rho_\pi \underbrace{\mathbb{P}_\pi(L = 1 | A = 1, S = 0)}_{\equiv T(\pi)} \\ + (1 - \rho_\pi)\mathbb{P}_\pi(L = 1 | A = 0, S = 0), \quad (2)$$

where $\rho_\pi$ is the proportion treated. The proportion of treated patients who receive no benefit, $T(\pi)$, is the *treatment risk* of $\pi$. It represents wasted resources and, possibly, unnecessary exposure to side-effects. Thus standard minimization of $R(\pi)$ can yield excess $T(\pi)$.

In contrast, we may not tolerate a treatment risk greater than $\tau \in (0, 1)$, in which case a treatment allocation $\pi$ that solves the constrained problem

$$\min_{\pi \in \Pi} R(\pi) \quad \text{subject to} \quad T(\pi) \leq \tau, \quad (3)$$

makes an explicit trade-off between risks (Wang et al., 2018; Doubleday et al., 2022; Kallus, 2022). For clinical decision policies, it is moreover important to restrict the learning of $\pi$ to a class of interpretable policies $\Pi$, e.g., rule-based policies (Rudin, 2019).

### 2.1. Data-Sampling Process

Neither risks in (3) are known and they are to be estimated using finite samples from *either* observational or randomized trial studies. We will assume that the treatment effects, described by $p(L|A, X, U)$ in (1), remain invariant.

For *observational* studies of the patient population, the causal structure of the sampling process is described in Figure 2b. The joint distribution of the process admits a causal factorization

$$p(X, U, A, L | S = 0) = \\ p(L|A, X, U) \, p(A|X, U) \, p(X, U | S = 0), \quad (4)$$

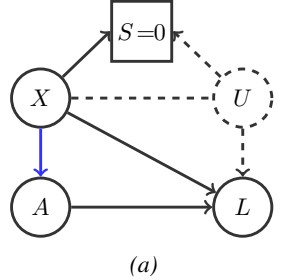 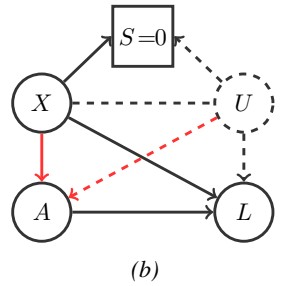 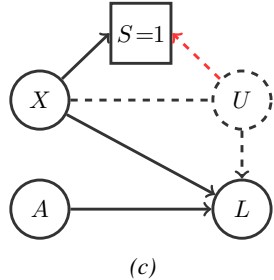

*(a)*             *(b)*             *(c)*

*Figure 2.* Structural causal models, specified by acyclic directed graphs. Causal structure of the decision process under policy $\pi$ in (a), where only observed covariates $X$ can influence treatment decisions $A$ (blue arrow). Causal structures of data-generating processes in (b) observational studies and (c) randomized trials. In (b), both observed covariates $X$ and unobserved factors $U$ may jointly influence treatment decisions $A$ and health loss $L$, introducing unmeasured confounding. In (c), unobserved factors $U$ may influence selection into trials, introducing unmeasured selection factors. The (conditional) indicator $S$ determines inclusion in a randomized trial study, cf. (Westreich, 2019).

where the past allocation of treatments, described by $p(A|X, U)$, differs from (1) and may depend on $U$.

In *trial* studies, by contrast, treatment allocation is randomized but individuals are selected into a study (indicated by $S = 1$). The causal structure of the sampling process is described in Figure 2c. The joint distribution of the process admits a causal factorization

$$
\begin{aligned}
p(X, U, A, L | S = 1) = \\
p(L | A, X, U) \, p(A) \, {\color{red} p(X, U | S = 1)},
\end{aligned}
\tag{5}
$$

where the randomized allocation $p(A)$ is known, but the trial population has characteristics that differ systematically from the intended patient population, as indicated by $p(X, U | S = 1)$. Thus, $X$ and $U$ may include factors that select individuals into trials.

The population and treatment risks in (2) are point identifiable if one could sample the joint distribution (4) *or* (5) (Peters et al., 2017; Westreich et al., 2017). However, the problem we face in *either* case is that we can *only* sample $(X_i, A_i, L_i)$ from the observable *marginals* $p(X, A, L | S = s)$ of (4) and (5), respectively. Therefore, the possibility of *unobserved* factors $U$, which is in general untestable, threatens the validity of any inferences drawn from observational or trial studies (Wasserman, 2013).

### 2.2. Degrees of Miscalibration

To estimate the population and treatment risks in (3) from $p(X, A, L | S = s)$, we use models of the past treatment assignment or trial selection process.

For *observational* studies, we learn a treatment assignment model $\widehat{p}(A|X)$. We then employ the approach of Tan (2006) and consider the nominal odds of assigning treat-

ment $A$ to be miscalibrated by a factor of at most $\Gamma_a$

$$
\frac{1}{\Gamma_a} \leq \underbrace{\frac{1 - p(A|X, U)}{p(A|X, U)}}_{\text{unknown odds}} \Bigg/ \underbrace{\frac{1 - \widehat{p}(A|X)}{\widehat{p}(A|X)}}_{\text{nominal odds}} \leq \Gamma_a \text{ (a.s.)}.
\tag{6}
$$

For *randomized trial* studies, we learn a model for the selection of individuals into the trial $\widehat{p}(S|X)$, which can be obtained using only covariate data from the trial and patient populations (Westreich, 2019). We consider the nominal selection odds to be miscalibrated by a factor of at most $\Gamma_s$

$$
\frac{1}{\Gamma_s} \leq \underbrace{\frac{p(S = 0|X, U)}{p(S = 1|X, U)}}_{\text{unknown odds}} \Bigg/ \underbrace{\frac{\widehat{p}(S = 0|X)}{\widehat{p}(S = 1|X)}}_{\text{nominal odds}} \leq \Gamma_s \text{ (a.s.)}.
\tag{7}
$$

We refer to $\Gamma \geq 1$ as the *degree of odds miscalibration*. A standard means of benchmarking a range of credible values for $\Gamma$ is to remove individual covariates $X_j$ and treat them as unobserved $U$ to obtain odds ratios as above (Manski, 2007; Ichino et al., 2008; Huang, 2024; Ek & Zachariah, 2024). We demonstrate this approach in Section A.2. As will be shown in Lemma 4.1 below, under either assumption (6) or (7) the risks in (2) are undetermined but can assume a range of possible values. That is, both $R(\pi)$ and $T(\pi)$ are *partially identifiable*.

### 2.3. Learning under Partial Identifiability

We now turn (3) into a learning problem using finite samples $\mathcal{D} = \{(X_i, A_i, L_i)\}$ drawn independently and identically from $p(X, A, L | S = s)$. For any specified degree of odds miscalibration $\Gamma \geq 1$, we seek a policy $\pi$ that approximately minimizes the population risk $R(\pi)$ while certifying that its treatment risk $T(\pi)$ will not exceed a specified tolerance $\tau$, with a high probability. That is, the learning method should map $\mathcal{D}$ to a policy in $\Pi$ that satisfies

$$
\boxed{\mathbb{P}\big(T(\pi) \leq \tau \mid S = s\big) \geq 1 - \alpha}
\tag{8}
$$

for all degrees of odds miscalibration up to $\Gamma$, where $\alpha$ is the tolerated probability of failure. Such a learning method would certify that treatment allocations control the proportion of treated who receive no benefit to be no more than $\tau$.

*Remark* 2.1. While the causal structures for observational and randomized trial studies represent the most common scenarios, our framework also handles a *third* case in which observational studies are conducted with respect to a study population that may differ from the patient population.

## 3. Background

Treatment effect estimation has traditionally focused on the average treatment effects (ATE) across populations, typically using data from randomized controlled trials (Imbens & Rubin, 2015). Optimal treatment allocation is instead focused on individual-level heterogeneity to assign treatments to those are most likely to benefit, see for example; Manski (2004); Dudík et al. (2011); Qian & Murphy (2011); Zhang et al. (2012); Zhao et al. (2012); Swaminathan & Joachims (2015); Athey & Imbens (2016); Wager & Athey (2018); Nie & Wager (2021), or the overview in Hoogland et al. (2021) for the case of binary outcomes. An important recent extension of this problem formulation is policy learning with continuous health outcomes under partial identifiability (Kallus & Zhou, 2021; Cui, 2021; Christensen et al., 2023; Adjaho & Christensen, 2023; Yata, 2025; Ben-Michael et al., 2025).

Another related line of work is concerned with policy learning under constraints, assuming point identifiability of risks (Kitagawa & Tetenov, 2018; Athey & Wager, 2021). Specifically, Wang et al. (2018) and Doubleday et al. (2022) consider problems in which a primary health outcome and a secondary adverse health variable are observed (both continuous). They learn linear or decision-rule policies that maximize treatment benefit while constraining the expected adverse health outcome, i.e., secondary harm. The proposed learning methods are, however, not certified to control the out-of-sample harm. In contrast, the method developed herein can readily control secondary harms with confidence, using a secondary loss $\widetilde{L} \in \{0, 1\}$ that indicates a side effect.

A different notion of harm was defined in Kallus (2022). For each patient, consider *counterfactually* applying $A = 0$ and $A = 1$ simultaneously, so that there are two corresponding *potential* health outcomes $(L(0), L(1))$, each indicating non-beneficial outcomes. The counterfactual harm of $\pi$ is then defined as the fraction negatively affected by the treatment: $\mathbb{E}[\pi(X)\mathbb{P}(L(0) = 0, L(1) = 1|X)]$. While this fraction is unobservable, a novel bound and asymptotically valid confidence interval for it was derived by Kallus (2022). The bound was tightened using additional assump-

tions by Li et al. (2023), who employed it in a constrained policy learning method that achieves control of counterfactual harm but only asymptotically. While philosophically appealing, this notion of harm is unverifiable since counterfactuals $(L(0), L(1))$ can never be simultaneously observed (cf. Sarvet & Stensrud (2023)). In contrast, the control over the proportion of treated who receive no benefit, i.e., $T(\pi) \leq \tau$, can be validated. It is therefore of value in clinical practice and can help guide decision-makers to strike trade-offs in the use of finite resources to improve health outcomes.

Simple treatment policies play an important role in ensuring effective, transparent, and scalable care. More complex data-driven models can often optimize treatment decisions, but tend to lack interpretability and, importantly, may be difficult to implement in practice (Rudin, 2019; Athey & Wager, 2021). Simple policies, on the other hand, enhance clinical interpretability, allowing healthcare providers to understand and apply treatment recommendations with confidence (Kitagawa & Tetenov, 2018). This is particularly important in high-stakes medical settings where decisions must be made reliably without delay (Caruana et al., 2015). To the best of our knowledge, problems of overtreatment and 'low-value' treatment allocations (Brownlee et al., 2017; Lyu et al., 2017; Ooi, 2020) have remained unexplored in the literature on policy learning.

In standard prediction problems, Bates et al. (2021) construct prediction sets that provide finite-sample guarantees on coverage probability without requiring strong distributional assumptions about the data. We extend this technique to the partially identified policy learning problem so as to derive guarantees on controlling the treatment risk, even in the case of confounding or selection bias.

## 4. Method

We will now propose a method to learn a policy that satisfies (8) with a user-specified treatment risk tolerance $\tau$ while minimizing the population risk. Since the risks in (2) are not point identifiable when sampling from $p(X, A, L|S = s)$, we first derive their upper bounds for any given degree of odds miscalibration $\Gamma \geq 1$.

*Lemma* 4.1. The population and treatment risks are upper bounded by

$$R(\pi) \leq \mathbb{E}\left[L \cdot \overline{W}^{\Gamma}\big|S = s\right] \quad \text{and}$$
$$T(\pi) \leq \mathbb{E}\left[L \cdot \frac{\mathbb{1}(A = 1)}{p_\pi(A = 1|S = 0)} \cdot \overline{W}^{\Gamma}\big|S = s\right], \quad (9)$$

where $\overline{W}^{\Gamma}$ denote weights. These are given by

$$\overline{W}^{\Gamma} = \left[1 + \Gamma_a\big(\widehat{p}(A|X)^{-1} - 1\big)\right] \cdot \mathbb{1}(A = \pi(X)), \quad (10)$$

in the case of observational data ($S = 0$), and

$$\overline{W}^\Gamma = \Gamma_s \cdot \frac{\widehat{p}(S=0|X)}{\widehat{p}(S=1|X)} \cdot \frac{p(S=1)}{p(S=0)} \cdot \frac{\mathbb{1}(A = \pi(X))}{p(A|X)}, \quad (11)$$

when using randomized data ($S = 1$). Moreover, if $\Gamma = 1$, then (9) holds with equalities.

*Remark* 4.2. Since the policy $\pi$ only takes $X$ as an input, its treatment probability $p_\pi(A = 1|S = 0) = \int \mathbb{1}(1 = \pi(x)) \, p(x|S = s)dx$ is point identifiable.

The proofs of the results presented here appear at the end of the section.

Now consider using upper bounds (9) for the risks in problem (3). To achieve risk control (8), we apply sampling splitting by randomly splitting $\mathcal{D}$ into two parts $\mathcal{D}_m$, and $\mathcal{D}_n$ with $m$ and $n$ samples, respectively. The first split $\mathcal{D}_m$ is used to learn a *nominal* policy.

*Definition* 4.3. Let $\widehat{\mathbb{E}}_m[Z] = \frac{1}{m}\sum_{i=1}^m Z_i$ denote the empirical mean. The *nominal policy* using $\mathcal{D}_m$ is defined as

$$\pi(X; t) = \arg\min_{\pi \in \Pi} \widehat{\mathbb{E}}_m \left[ L \cdot \overline{W}^\Gamma \right]$$

$$\text{s.t. } \widehat{\mathbb{E}}_m \left[ L \cdot \frac{\mathbb{1}(A = 1)}{p_\pi(A=1|S=0)} \cdot \overline{W}^\Gamma \right] \leq t, \quad (12)$$

where $t \in (0, 1)$ is a nominal tolerance parameter. Thus (12) is a mapping from $(\mathcal{D}_m, t, \Gamma)$ to $\Pi$.

Thus comparing with (9), the nominal policy (12) aims to minimize the empirical bound on the population risk subject to a constraint. We now show that this constraint can be made to achieve the desired (8).

The unknown treatment risk of (12) is upper bounded by (9), which can be expressed as

$$\overline{T}(t) = \mathbb{E}[V(t)|S = s],$$

where

$$V(t) = L \cdot \frac{\mathbb{1}(A = 1)}{p_\pi(A = 1|S = 0)} \cdot \overline{W}^\Gamma(t).$$

The second dataset $\mathcal{D}_n$ is now used to construct an upper *confidence bound*, denoted $\overline{T}_n^\alpha(t)$, such that

$$\mathbb{P}(\overline{T}(t) \leq \overline{T}_n^\alpha(t)|S = s) \geq 1 - \alpha. \quad (13)$$

*Theorem* 4.4. Let $\overline{T}_n^\alpha(t)$ denote an upper bound that satisfies (13). For any specified treatment risk tolerance $\tau$, define the empirical tolerance

$$\tau_n = \arg\min_{t\in(0,1)} \widehat{\mathbb{E}}_n \left[ L \cdot \overline{W}^\Gamma(t) \right]$$

$$\text{s.t. } \tau > \overline{T}_n^\alpha(t'), \quad \forall t' \leq t. \quad (14)$$

If the nominal $\pi(X; \tau)$ obtained from (12) achieves the constraint with equality, then $\pi(X; \tau_n)$ is certified to control the treatment risk according to (8). That is, $\pi(X; \tau_n)$ satisfies $T(\pi) \leq \tau$ with probability no less than $1 - \alpha$ under all degrees of miscalibration up to $\Gamma$.

*Remark* 4.5. There are several possible confidence bounds (13). One method is based on Bentkus inequality which becomes tight for binary data (Bentkus, 2004). It is obtained as follows: Let $V_{\max}$ denote an upper limit on $V(t)$ which contains the binary $L$. We therefore expect $V(t)$ to be either 0 or clustered closer around some values towards $V_{\max}$. A confidence bound $\overline{T}_n^\alpha(t)$ that is tight for a binary loss is useful in this case. (With a smaller variance, other bounds may be tighter, see (Bates et al., 2021).) Define

$$g(a; \overline{T}(t)) = \min\left( \exp\{ -nh(a; \overline{T}(t))\}, \right.$$
$$\left. e \cdot \text{CDF}(\lceil na \rceil; n, \overline{T}(t)) \right),$$

where

$$h(a; \overline{T}) = a \log(a/\overline{T}) + (1 - a)\log((1 - a)/(1 - \overline{T})),$$

and $\text{CDF}(\cdot; n, p)$ is the cumulative distribution function of a binomial random variable with sample size $n$ and success probability $p$. Then

$$\overline{T}_n^\alpha(t) = \sup \left\{ \overline{T} : g\left( \frac{\widehat{\mathbb{E}}_n[V(t)]}{V_{\max}}; \frac{\overline{T}}{V_{\max}} \right) \geq \alpha \right\}, \quad (15)$$

is a valid upper confidence bound (13).

*Remark* 4.6. The empirical tolerance $\tau_n$ in (14) is chosen to yield the lowest bound on the in-sample population risk while satisfying the constraint with respect to the upper confidence bound.

*Remark* 4.7. The complete method is summarized in Algorithm 1 and an implementation together with experimental results are available here.

---

**Algorithm 1** Learn policy $\pi(X)$

---

**input** Data $\mathcal{D}$, degree of miscalibration $\Gamma$, policy class $\Pi$, parameter $\tau$, confidence level $1 - \alpha$.

**output** Policy $\pi(X)$

1: Randomly split $\mathcal{D}$ into $\mathcal{D}_m$, and $\mathcal{D}_n$.
2: **for** $t \in (0, 1)$ **do**
3:     Learn $\pi(X; t)$ as in (12) using $\mathcal{D}_m$.
4:     Save $\pi(X; t)$.
5: **end for**
6: **for all** saved $\pi(X; t)$ **do**
7:     Compute upper bound $\overline{T}_n^\alpha(t)$ from (13), e.g., via (15), using $\mathcal{D}_n$.
8: **end for**
9: Select $\tau_n$ as in Equation (14) using $\mathcal{D}_n$.

---

*Proof of Theorem 4.1.* We start by proving the bound for $T(\pi)$. Since $L \in \{0, 1\}$, we have that

$$
\begin{aligned}
T(\pi) &= \mathbb{P}_\pi(L = 1 | A = 1, S = 0) \\
&= \mathbb{E}_\pi[L | A = 1, S = 0] \\
&= \sum_\ell \ell \cdot p_\pi(\ell | A = 1, S = 0),
\end{aligned}
$$

and, moreover, using the chain rule: $p_\pi(\ell | A = 1, S = 0) = p_\pi(\ell, A = 1 | S = 0) / p_\pi(A = 1 | S = 0)$, it follows that

$$
\begin{aligned}
T(\pi) &= \sum_\ell \sum_a \ell \frac{p_\pi(\ell, a | S = 0)}{p_\pi(A = 1 | S = 0)} \mathbb{1}(a = 1) \\
&= \mathbb{E}_\pi \left[ L \frac{\mathbb{1}(A = 1)}{p_\pi(A = 1 | S = 0)} \Big| S = 0 \right].
\end{aligned}
$$

Next, we note that

$$
\begin{aligned}
\mathbb{E}_\pi[Z | S = 0] &= \mathbb{E} \left[ Z \cdot \frac{p_\pi(X, U, A, L | S = 0)}{p(X, U, A, L | S = s)} \Big| S = s \right] \\
&= \mathbb{E}[Z \cdot W_\pi | S = s],
\end{aligned}
$$

where $W_\pi = p_\pi(X, U, A, L | S = 0) / p(X, U, A, L | S = s)$ is an importance weight given by (4) or (5), depending on whether we are using observational or trial data. The weight is upper bounded $W_\pi \le \overline{W}^\Gamma$ from (10) or (11), using (6) or (7) correspondingly. This proves the upper bound for $T(\pi)$. The bound for $R(\pi)$ is analogous. $\square$

*Proof of Theorem 4.4.* For notational convenience, we drop the symbol of conditioning on $S = s$ in the expressions that follow. We want to bound the probability of the event

$$
\overline{T}(\tau_n) > \tau \tag{16}
$$

to ensure it is no greater than $\alpha$.

For policy $\pi(X; t)$, which satisfies the constraint in (12), $\widehat{\mathbb{E}}_m[V(t)] = \frac{1}{m} \sum_{i=1}^m V_i(t) \le t$ so that after applying an expectation on both sides of the inequality we have

$$
t \ge \frac{1}{m} \sum_{i=1}^m \mathbb{E}[V_i(t)] = \mathbb{E}[V(t)] = \overline{T}(t). \tag{17}
$$

Using this relation, it follows that

$$
\tau_n \ge \overline{T}(\tau_n) > \tau \ge \overline{T}(\tau) \tag{18}
$$

for the event (16) under consideration.

By construction of the empirical tolerance $\tau_n$ in (14), we have that

$$
\forall t \le \tau_n \; : \; \tau > \overline{T}_n^\alpha(t). \tag{19}
$$

From (18), $\tau_n > \tau$ and by (19) we have $\tau > \overline{T}_n^\alpha(\tau)$. Since $\pi(X; \tau)$ yields $\overline{T}(\tau) = \mathbb{E}\left[ \widehat{\mathbb{E}}_m[V(\tau)] \right] = \tau$ via (17), it follows that

$$
\overline{T}(\tau) > \overline{T}_n^\alpha(\tau). \tag{20}
$$

However, from (13), we have that

$$
\mathbb{P}(\overline{T}(\tau) > \overline{T}_n^\alpha(\tau)) \le \alpha. \tag{21}
$$

That is, the event occurs with a probability of at most $\alpha$. The proof technique is similar to that employed in (Bates et al., 2021) for the error control of prediction sets. $\square$

# 5. Experimental

To illustrate the properties of the proposed method of treatment risk control, we use both synthetic and real-world datasets. For concreteness, we consider the policy class $\Pi$ in (12) to be a family of fast-and-frugal decision trees (Gigerenzer et al., 2000). The learning problem is solved following the greedy approach proposed in Zhang et al. (2015) with the key difference that the constraint in (12) is also evaluated in each search step. If the tolerance $\tau$ becomes too small to satisfy, the method simply returns the baseline policy $\pi(X) \equiv 0$. Further details are provided in Section A.1.

## 5.1. Synthetic Data

We begin by describing an observational data distribution (4). In the first case, there is no $U$ that can affect both treatment and health outcomes. The covariates $X$ are two-dimensional and $p(X | S = 0)$ is given by

$$
X = \begin{bmatrix} X_1 \\ X_2 \end{bmatrix} | S = 0 \sim \mathcal{U}(30, 80)^2. \tag{22}
$$

The assignment of treatment actions $A \in \{0, 1\}$ follows a known distribution

$$
p(A = 1 | X) = \sigma \left( 0.5 - \frac{X_1 - 30}{50} \right), \tag{23}
$$

where $\sigma(x) = (1 + \exp(-x))^{-1}$. The health loss probability for both treatment options follows

$$
\begin{aligned}
&p(L = 1 | X, A = 0) = 0.8 \quad \text{and} \\
&p(L = 1 | X, A = 1) = 0.01 \cdot (X_1 - 30).
\end{aligned} \tag{24}
$$

We generate $|\mathcal{D}| = m + n = 1000 + 1000$ samples. The set $\mathcal{D}_m$ is used to learn $\pi(X; t)$ in (12), where $t \in (0, 0.5]$ is evaluated at 200 equally spaced points. The decision tree is restricted to a single split, where the covariates are discretized into 200 bins. The other set $\mathcal{D}_n$ is used to form $\tau_n$ in (14) with a specified $\tau$ and confidence level $1 - \alpha = 90\%$. The resulting policy is $\pi(X; \tau_n)$.

We evaluate the population and treatment risks for 1000 policies learned from different draws of $\mathcal{D}$. Initially, we assume no miscalibration of the treatment assignment odds ($\Gamma = 1$). In Figure 1a, we see that the treatment risk is effectively controlled by $\tau$, with a probability of at least

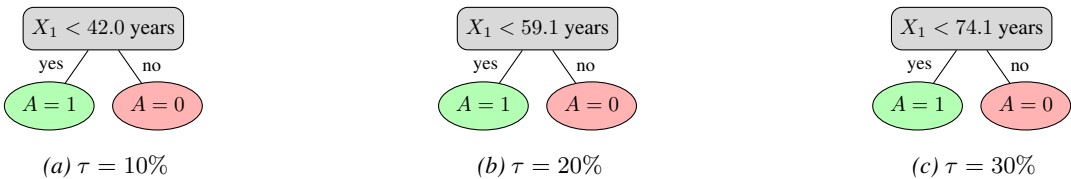

*Figure 3.* Treatment allocation policies $\pi(X; \tau_n)$ learned from a synthetic dataset under different treatment risk tolerances $\tau$. Here $\Gamma = 1$.

$1 - \alpha$. In contrast with Figure 1b, the trade-off between population and treatment risks is visible. (Additional comparisons with the nominal policy (12) and policies that assume odds miscalibration up to $\Gamma = 2$ are provided in Section A.3.) For illustration purposes, the certified learned policies $\pi(X; \tau_n)$ for three different risk tolerances $\tau$ are visualized in Figure 3. As expected, a lower tolerance $\tau$ results in treatments being assigned to individuals with lower $X_1$. As $\tau$ increases, the proportion of treated individuals also increases.

We now modify the data-generating process to include an unmeasured confounding variable $U$, uniformly distributed $U \sim \mathcal{U}(0, 1)$. The treatment assignment probability is set to

$$
\begin{aligned}
p(A = 1 | X, U) = {} & \mathbb{1}(U < 0.5) \cdot \frac{p_{\text{nom}}(X)}{2 - p_{\text{nom}}(X)} \\
& + \mathbb{1}(U \geq 0.5) \cdot \frac{p_{\text{nom}}(X)}{0.5 + 0.5 p_{\text{nom}}(X)},
\end{aligned}
$$

where $p_{\text{nom}}(X)$ is the nominal probability model in (23). The loss for the treated is drawn from

$$
\begin{aligned}
p(L = 1 | A = 1, X, U) = {} & \mathbb{1}(U < 0.5) \cdot 0.02 \cdot (X_1 - 30) \\
& + \mathbb{1}(U \geq 0.5) \cdot 0.002 \cdot (X_1 - 30).
\end{aligned}
\tag{25}
$$

We assume that the learning method uses $\widehat{p}(A|X) = p_{\text{nom}}(X)$ as a nominal model, which corresponds to an odds miscalibration degree of $\Gamma = 2$ in (6). We repeat the evaluation of the learned policies with $1 \leq \Gamma \leq 2$. In Figure 4a, it is evident that the treatment risk cannot be controlled when assuming no miscalibration ($\Gamma = 1$). In contrast, if nominal odds could be off by a factor of up to $\Gamma = 2$, the treatment risk is indeed controlled. In Figure 4b, the trade-off between treatment and population risks is still visible, and accommodating for $\Gamma = 2$ yields a tighter treatment risk constraint and consequently higher population risk. Additional results in the case of randomized trial data are included in Section A.3.

### 5.2. STAR Data

For illustration, we also test our method on real-world data from the Tennessee Student/Teacher Achievement Ratio (STAR) study (Achilles et al., 2008; Krueger, 1999), a ran-

domized controlled trial on class size conducted between 1985 and 1989. In this study, preschool through third-grade students and their teachers were randomly assigned to one of three class types: small (1317 students per teacher), regular (2225 students per teacher) and regular with an aide (2225 students per teacher plus a full-time assistant). However, in our analysis, we focus only on the first two groups. Each student is characterized by $X$, which covers 11 covariates such as birth month, gender, teacher career, and experience. Additional details can be found in Section A.4.

Following Kallus et al. (2018), we define first-grade class type as the treatment, as many students were not enrolled in the study during preschool ($A = 0$ is 'regular class size' and $A = 1$ is 'small class'). The outcome variable $L$ is the achievement test score at the end of first grade, calculated as the sum of the standardized math, reading and listening scores. This score is then binarized, with $L = 0$ indicating a sum above the median and $L = 1$ indicating a sum at or below the median. Students who were not part of the STAR study in first grade, had missing outcome data or were assigned to the regular class with an aide were excluded.

The final dataset consists of 4218 students. A 50 percent of the samples is randomly split for policy construction in (12), where $t$ is evaluated at 100 equally spaced points in the range $(0, 0.8]$. The decision tree is constrained to a maximum of three splits, and similarly to the synthetic case, continuous covariates are discretized into 200 bins. The next 25 percent of samples are used to form (14) for the certified policy $\pi(X; \tau_n)$. The remaining 25 percent of samples is used for evaluation. For the STAR data, a full evaluation, as conducted in the synthetic case, is not feasible. Instead, we use 100 random splits to evaluate the policies. Assuming no miscalibration ($\Gamma = 1$), the resulting policies show valid coverage for all values of $\tau$ (Figure 5a). In this scenario, the trade-off between controlling the treatment risk and minimizing the population risk still remains evident (Figure 5b). In Figure 6, the learned policies for different tolerances $\tau$ are visualized. A low $\tau$ results in no assigned treatment, while a high $\tau$ leads to a treat-all policy. For $\tau = 0.35$ and $\tau = 0.40$ policy trees of depth two are identified and a portion of the population receives treatment.

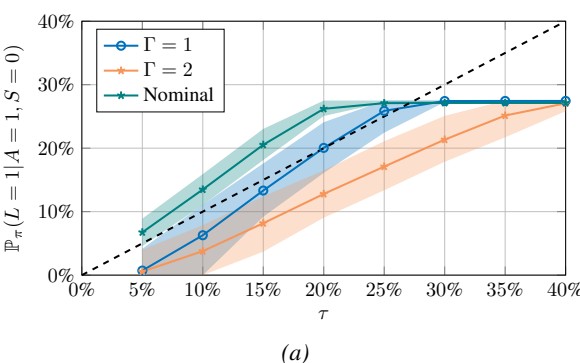
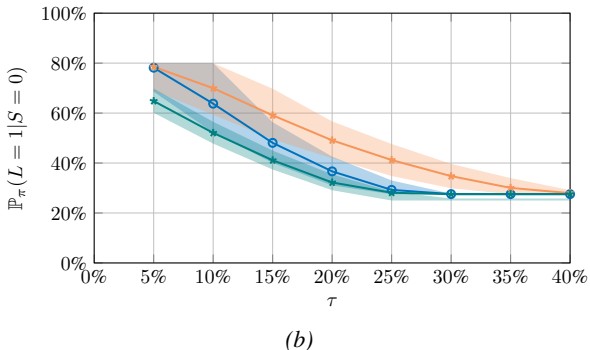

*(a)*                *(b)*

*Figure 4.* Treatment risk (a) and the population risk (b) under $\pi$ versus tolerance $\tau$. The risks are only partially identifiable due to unmeasured confounding, and policies $\pi(X; \tau_n)$ are certified to control the treatment risk with probability of at least $1 - \alpha = 90\%$, up to a specified degree of odds miscalibration $\Gamma$. The nominal policy $\pi(X; \tau)$ in (12) is included as a baseline, where $\pi(X; 1)$ using $\Gamma = 1$ is obtained by standard empirical risk minimization. The shaded regions represent the range between the 10th and 90th percentiles of policies learned using 1,000 different datasets. (a) For an assumed degree of odds miscalibration of $\Gamma = 2$, the treatment risk is controlled as expected. Assuming no miscalibration, $\Gamma = 1$, is less credible and we also see there is no risk control in this case. (b) The corresponding population risks.

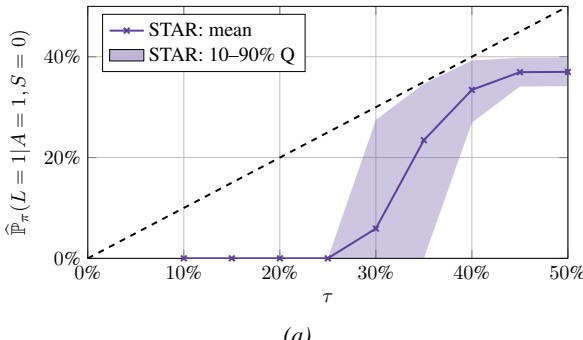
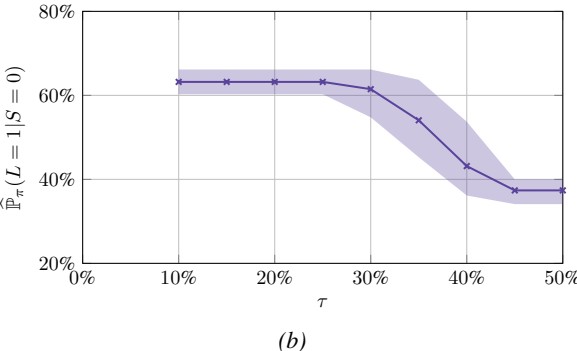

*(a)*                *(b)*

*Figure 5.* Treatment and population risks of policies $\pi(X; \tau_n)$ learned from STAR dataset. The shaded regions are obtained by randomized sample splitting.

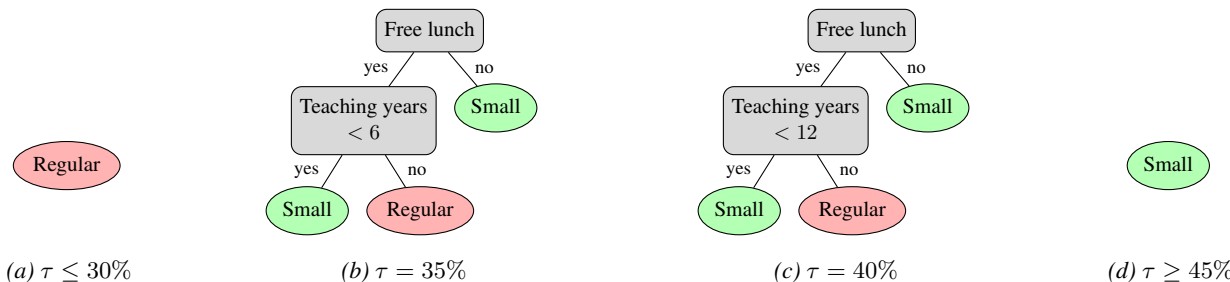

*(a)* $\tau \leq 30\%$       *(b)* $\tau = 35\%$       *(c)* $\tau = 40\%$       *(d)* $\tau \geq 45\%$

*Figure 6.* Examples of learned treatment policies $\pi(X; \tau_n)$ using the STAR dataset, with student covariates $X$ in grey, for different risk tolerances $\tau$.

## 6. Discussion

We introduced a learning method that advances trustworthiness in data-driven decision-making by achieving certified control of the treatment risk while seeking beneficial treatment allocations. The proposed approach achieves this control in finite samples, even in partially identified settings up to any specified degree of odds miscalibration. While the methodology is motivated by the need to reduce overtreatment and low-value care in clinical settings (Brownlee et al., 2017; Lyu et al., 2017; Ooi, 2020), we believe it can provide a valuable tool for high-stakes applications such as personalized medicine, economic policy, and safety-critical systems.

## Acknowledgements

This work was partially supported by the Swedish Research Council (contract number 2024-03903) and the Wallenberg AI, Autonomous Systems and Software Program (WASP) funded by the Knut and Alice Wallenberg Foundation.

## Impact Statement

The proposed method provides a principled way to control treatment risk on average across a given population. This can provide a tool to control wasteful treatment allocations, reduce secondary harms from overtreatments, and improve the value of care in health systems.

However, in settings where certain subgroups, defined by sensitive or high-stakes features such as age or socioeconomic status, exhibit significantly different risk profiles, (average) risk control may be insufficient. In such cases, it may be necessary to stratify the population into appropriate subpopulations and apply risk control separately within each group to ensure fairness.

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

## A. Additional Details Experiments

All experiments were carried out on a laptop using only the CPU. The synthetic experiments required approximately two and a half to three hours for 1000 runs, while the real-data experiments took 10 to 15 minutes for 100 runs.

### A.1. Fast-and-Frugal Policy Learning

Fast-and-frugal trees (FFTs) are rule-based decision trees designed for high-stakes, time-sensitive environments such as medical diagnostics and emergency response (Gigerenzer et al., 2000). Their structure supports early decision-making, enabling some cases to be resolved without using all available information. For example, in medical testing, certain tests may be unnecessary for some patients, helping to optimize both time and resources (Katsikopoulos et al., 2021).

We use FFTs to construct the policies in (12), following the greedy approach of Zhang et al. (2015). We restrict our rules to one variable in each condition and grid continuous covariates. The algorithm evaluates

$$\widehat{\mathbb{E}}_m \left[ L \cdot \overline{W}^\Gamma \right], \tag{26}$$

for each potential split and selects the one that minimizes this value. The key distinction from Zhang et al. (2015) is that we additionally evaluate

$$\widehat{\mathbb{E}}_m \left[ L \cdot \frac{\mathbb{1}(A = 1)}{p_\pi(A = 1|S = 0)} \cdot \overline{W}^\Gamma \right] \leq t, \tag{27}$$

at each greedy step and impose a constraint that only permits splits if (27) is satisfied. Before a split is performed, the current tree is stored. At each step, two new candidate trees are generated: one that continues to grow if the criterion is satisfied, and another that extends in the alternative direction. This process iterates until a stopping criterion is reached, either when the maximum depth is attained or when further splits fail to improve the objective function. Once all trees have been constructed, the final model is selected as the tree that minimizes the objective function.

### A.2. Benchmarking Degree of Miscalibration

We now demonstrate how a credible range for $\Gamma$ can be benchmarked, using an approach from Huang et al. (2021) and Ek & Zachariah (2024). To this end, we estimate the propensity score via logistic regression for the STAR dataset for illustration. As a first step, we assess whether this model offers sufficient flexibility for the task. The nominal assignment odds in (6) are discretized into five bins. Within each bin, the unknown assignment odds are estimated by computing the empirical ratio of the samples for which $A = 1$ and $A = 0$, respectively. If the model is sufficiently flexible, the estimated assignment odds should approximate the nominal odds within each bin. This is observed in Figure 7a.

Next, we continue with a benchmarking approach designed to account for the potential influence of unobserved individual factors $U$, by treating observed covariates in $X$ as unmeasured. Let the $k$th covariate $X_k$ correspond to an omitted factor, while $X_{-k}$ denotes all the remaining observed covariates, and define

$$\widehat{\text{odds}}(X_{-k}, X_k) = \frac{1 - \widehat{p}(A|X)}{\widehat{p}(A|X)}, \quad \widehat{\text{odds}}(X_{-k}) = \frac{1 - \widehat{p}(A|X_{-k})}{\widehat{p}(A|X_{-k})}. \tag{28}$$

The ratio $\widehat{\text{odds}}(X_{-k}, X_k)/\widehat{\text{odds}}(X_{-k})$ is then used to benchmark odds ratio in (6). Figure 7b illustrates this for the two most influential observed covariates. If the unobserved individual factor $U$ has an effect on the propensity score that is no greater than these, then a credible range for $\Gamma$ is between 1.5 and 1.7.

### A.3. Synthetic Data

To complement the experiments related to the case of no miscalibration ($\Gamma = 1$) in Figures 1 and 3, we compare with the nominal policy $\pi(X; \tau)$ and a policy $\pi(X; \tau_n)$ certified to accommodate odds miscalibration up to degree $\Gamma = 2$. The latter is naturally more conservative than assuming $\Gamma = 1$, but, importantly, its risk control remains valid; see Figure 8 and Figure 9. Meanwhile the nominal $\pi(X; \tau)$ cannot certify risk control with any given probability.

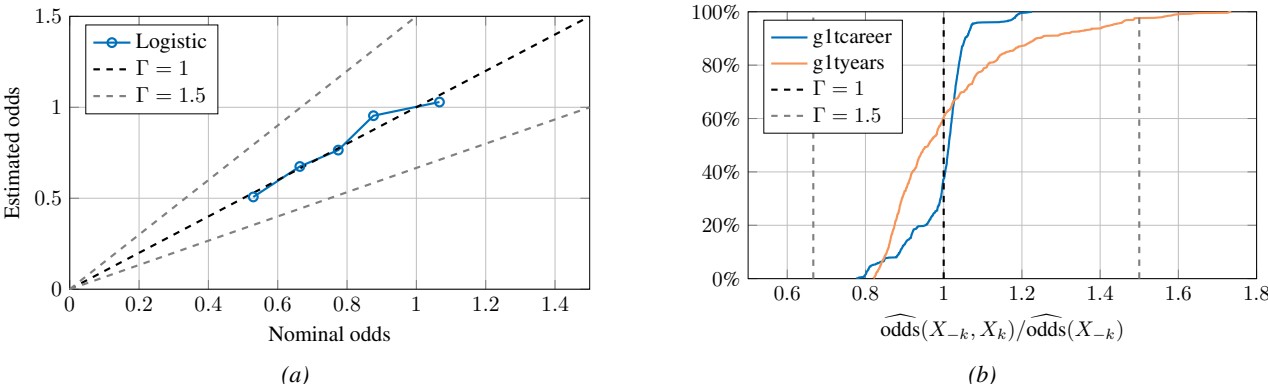

*Figure 7.* Benchmarking $\Gamma$. (a) Reliability diagram of the observed odds against the average predicted nominal odds. (b) Benchmarking $\Gamma$ using omitted covariates.

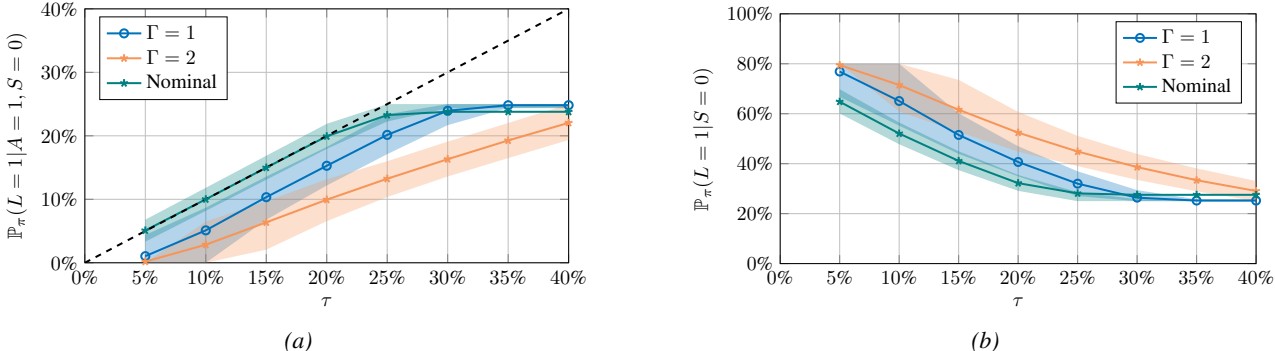

*Figure 8.* Treatment and poplution risks when they are point-identifiable from data (i.e., no miscalibration). Reproduces Figure 1 but also includes nominal policy $\pi(X; \tau)$ in (12) and a certified policy $\pi(X; \tau_n)$ that accommodates odds miscalibration up to $\Gamma = 2$.

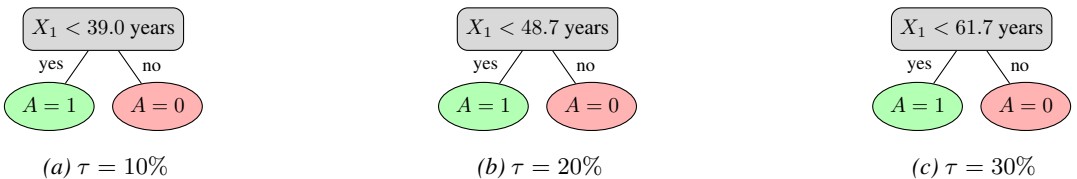

*Figure 9.* Treatment allocation policies $\pi(X; \tau_n)$ learned from a synthetic dataset. Same as Figure 3 but setting $\Gamma = 2$.

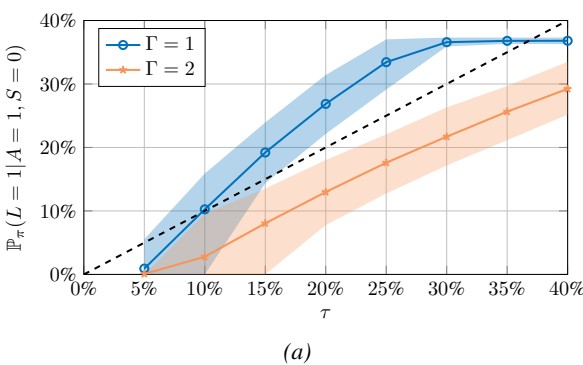

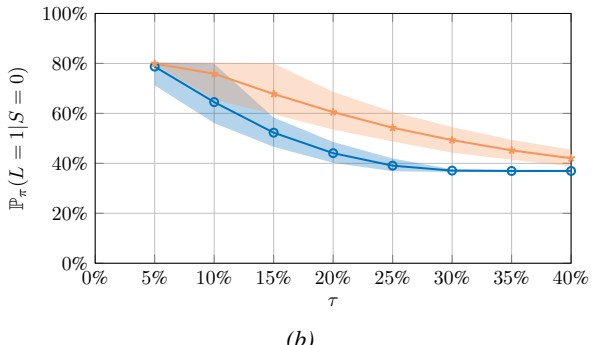

*(a)*          *(b)*

*Figure 10.* Treatment risk (a) and the population risk (b) under $\pi$ versus tolerance $\tau$. The risks are only partially identifiable due to unmeasured selection factors into randomized trials.

We also extend the synthetic experiments with experiments for RCT, similarly to the confounding case in the main paper. The covariates $X$ are two-dimensional and $p(X)$ is given by

$$X = \begin{bmatrix} X_1 \\ X_2 \end{bmatrix} \mid S = 1 \sim \mathcal{U}(30, 80)^2. \tag{29}$$

The assignment of treatment actions $A \in \{0, 1\}$ is $p(A = 1) = 0.5$ and is known. We also have a confounding variable $U$, uniformly distributed $U \sim \mathcal{U}(0, 1)$, that affects the sampling $S$ and the loss $L$. The sampling distribution is given by

$$p(S = 1 \mid X, U) = \mathbb{1}(U < 0.5) \cdot \frac{p_{\text{nom}}(X)}{2 - p_{\text{nom}}(X)} + \mathbb{1}(U \geq 0.5) \cdot \frac{p_{\text{nom}}(X)}{0.5 + 0.5 p_{\text{nom}}(X)},$$

where $p_{\text{nom}}(X)$ is the propensity score in (23). This corresponds to a miscalibration of $\Gamma = 2$ in (7). The loss for the treated is defined as in (25). The evaluation follows the procedure described in Section 5.1, with the key difference that, instead of relying on a closed-form expression, each Monte Carlo run is now evaluated using 18,000 additional samples drawn from the test distribution.

The decision policy $\pi(X; \tau_n)$ is learned for both $\Gamma = 1, 2$. In Figure 10a, we observe that for $\Gamma = 1$, the guarantee that the treatment risk remains below $\tau$ with a probability no less than $1 - \alpha$ is violated. In contrast, $\Gamma = 2$ provides a valid model for the selection odds, ensuring that the treatment risk of the resulting policies stays below $\tau$ in the guaranteed proportion of trials. Figure 10b illustrates the trade-off between treatment and population risks also when using randomized trial data. Using $\Gamma = 2$ tightens treatment risk control but increases the overall population risk as fewer receive treatment.

### A.4. STAR Data

For the experiments in Section 5.2 the 11 covariates in Table 1 are used.

### A.5. International Stroke Trial Data

The International Stroke Trial (IST) was a large randomized trial that evaluated the effects of aspirin and heparin in acute ischemic stroke. The original trial included 19,435 patients and compared four arms: aspirin, heparin, both, or none (Group et al., 1997). For this analysis, we only compare aspirin ($A = 1$) and no aspirin ($A = 0$) and heparin is viewed as a covariate. In total, our analysis included 23 covariates $X$, such as age, sex, level of consciousness, and neurological symptoms. The public data set and descriptions of the covariates are available in Sandercock et al. (2011). The outcome $L$ of interest is death at six-month follow-up. We exclude 984 patients from a preliminary study and patients with missing or unknown outcome data (153 patients), resulting in a final sample of 9154 patients in the non-treatment group and 9144 patients in the aspirin group.

For policy learning, we use the same setup as in Section 5.2. Assuming the weights are correctly assigned, the resulting policies show a valid coverage for all values of $\tau$ (Figure 11a). In this scenario the difference between treat and non-treatment is small (Figure 5b) and a majority of the resulting policies switch from treat non to treat all when $\tau$ is between 20% and 25%.

*Table 1.* The covariates used in the STAR experiments.

| Covariate | Description | Categorical |
|---|---|---|
| gender | Student gender | yes |
| race | Ethnicity | yes |
| g1promote | Promoted from grade 1 | yes |
| g1specin | Special instruction | yes |
| g1surban | School location | yes |
| g1freelunch | Free/reduced lunch | yes |
| birthmonth | Birth month | no |
| g1present | School days present | no |
| g1absent | School days absent | no |
| g1tcareer | Teachers career level | no |
| g1tyears | Teaching experience (years) | no |

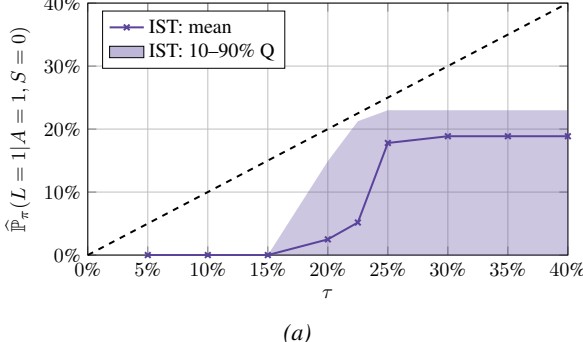

*(a)*

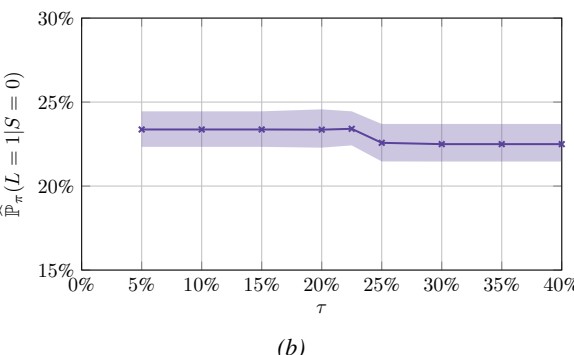

*(b)*

*Figure 11.* The estimated treatment risk (a) and population risk (b) under policy $\pi$ for different values of the treatment risk tolerance $\tau$ for the IST dataset.

