# OpenReview forum: "Learning Treatment Allocations with Risk Control Under Partial Identifiability"
_ICML.cc/2026/Conference — ICML 2026 regular_

### Official Review · Reviewer_XRtA · 2026-03-09

**Soundness:** 3
**Presentation:** 3
**Significance:** 2
**Originality:** 3
**Overall Recommendation:** 4
**Confidence:** 4

**Summary:**

The paper addresses the challenge of learning treatment allocation policies that control treatment risk (the proportion of treated patients who do not benefit) under partial identifiability due to unmeasured confounding or selection bias. The authors propose a method that leverages sample splitting and miscalibration bounds to provide finite-sample guarantees, ensuring the treatment risk remains below a specified tolerance while minimizing the overall population risk.

**Compliance With Llm Reviewing Policy:**

Affirmed.

**Final Justification:**

The authors addressed my concerns and I have no further comments. Thus, I'd stick to my current score 4 (which is positive).

**Key Questions For Authors:**

· Theoretical Formulation and Sample Size: The theoretical derivations heavily utilize empirical expectations $\hat{\mathbb{E}}$ directly. Would it be more rigorous to ground the core theory using true population expectations $\mathbb{E}$, and subsequently dedicate empirical analysis to the impact of finite dataset sizes? Specifically, an ablation on the sizes and ratios of $\mathcal{D}_m$ and $\mathcal{D}_n$ would be highly informative.
· Discontinuous Risk Curves: In the STAR dataset experiments, the treatment and population risks do not scale proportionally; instead, there are sudden, step-like changes around $\tau = 25\%$ and $\tau = 35\%$. Is this non-smooth behavior primarily an artifact of the strict structural limitations of the fast-and-frugal decision trees?
· Observational vs. Randomized Data: The theoretical sections clearly delineate between observational and randomized trial data structures. However, this distinction feels somewhat conflated or under-highlighted in the experimental section. Could you better clarify and contrast how the empirical performance differs between these two specific data-generating processes?

**Limitations:**

See Weaknesses and Questions.

**Strengths And Weaknesses:**

Strengths

· High Real-World Relevance: The motivation addresses a critical gap in precision medicine: the mitigation of overtreatment and low-value care. Framing this problem using a strict inequality constraint on treatment risk is a practical and meaningful contribution to the field.
· Rigorous and Elegant Formulation: The mathematical framework is exceptionally well-defined. The authors successfully and elegantly bridge terminology and formalisms from causal inference, off-policy learning (e.g., contextual bandits), and statistical machine learning.
· Solid Theoretical Grounding: The proposed policy learning paradigm directly translates from the underlying theory. The constrained optimization problem in Equation 12 is rigorously derived from the theoretical upper bounds established in Lemma 4.1

Weaknesses

· Flaws in the Policy Learning Paradigm: The empirical formulation in Equation 12 suffers from significant limitations regarding underlying optimization and generalizability. By relying on a hard empirical constraint that couples an indicator function $1(A=1)$ with inverse probability weights, the objective creates a highly non-convex and non-differentiable optimization landscape. This inherently precludes the use of standard gradient-based optimization frameworks. Consequently, the approach forces a reliance on computationally inefficient grid searches and greedy heuristics, as demonstrated by the authors' restriction to fast-and-frugal decision trees. This severely limits the paradigm's versatility, making it exceedingly difficult to scale or generalize to the more expressive, high-capacity model architectures (e.g., deep neural networks) that are standard in modern policy learning
· Underwhelming Empirical Trade-offs: The experimental results do not consistently demonstrate a compelling practical advantage. While treatment risk is effectively controlled at low $\tau$, performance degrades at higher thresholds, and the resulting population risk is often less than ideal. For example, at $\tau = 5\%$, the population risk reaches approximately 60%-70%. At $\tau = 40\%$, the population risk remains near 30% while treatment risk sits around 25%. It is debatable whether this strict control consistently outperforms models that simply ignore unmeasured confounding $U$.
· Insufficient Exploration of $\Gamma$: While the theoretical benchmarking of the miscalibration degree $\Gamma$ in Section 2.2 is elegant and intuitive , the experimental evaluation arbitrarily restricts this hyperparameter to $\Gamma = 1$ and $\Gamma = 2$. A finer-grained ablation study is required to fully understand the model's sensitivity and the precise trade-off dynamics driven by this parameter.
· Lack of Standard Unconstrained Baselines: The core novelty of the paper lies in constraining the treatment risk, which the authors claim is largely unexplored. Consequently, it is necessary to benchmark this against standard Empirical Risk Minimization (ERM) approaches that optimize for population risk without the treatment risk constraint, in order to explicitly quantify the cost of this constraint.

---

> ### Author Rebuttal · Authors · 2026-03-29
>
> We thank the reviewer for recognizing our contributions and providing constructive feedback.
>
> >Would it be more rigorous to ground the core theory using true population expectations $\mathbb{E}$, and subsequently dedicate empirical analysis to the impact of finite dataset sizes? Specifically, an ablation on the sizes and ratios of $\mathcal{D}_m$ and $\mathcal{D}_n$ would be highly informative.
>
> To avoid risking misunderstandings, let us restate how the core of the theory is grounded:
> 1. The key quantities $R(\pi)$ and $T(\pi)$ in Eq. (2) are population-based quantities defined using $\mathbb{E}$.
> 2. We seek to establish finite-sample valid control of $T(\pi)$ as defined in (8).
> 3. The route by which we establish the control over control of $T(\pi)$ uses empirical risk.
>
> In sum, there is no loss in rigor in the risk control via the use of empirical risk.
>
> The reviewer however also raises the interesting question of how the split and sizes of $m$ and $n$ impacts the learned policies. From the theoretical results, we can assert that treatment risk control over $T(\pi)$ remains unchanged and, all else equal, a larger $\mathcal{D}_n$ means that the empirical tolerance in (14) will decrease towards $\tau$. This reduces the conservativeness of the constraint for $\pi(X; \tau_n)$ and will thus tend to improve the population risk (via minimization of the empirical risk).
>
> Similarly, increasing $\mathcal{D}_m$ would reduce both population risk and treatment risks (up to the limits set by the process and policy class). A study of how the ratio between $m$ and $n$ would affect the results in a given example with a fixed sample budget is something we could explore further.
>
> >In the STAR dataset experiments, the treatment and population risks do not scale proportionally; instead, there are sudden, step-like changes around  and . Is this non-smooth behavior primarily an artifact of the strict structural limitations of the fast-and-frugal decision trees?
>
> We have been unable to run extensive tests with other interpretable policy classes, but our experiments and experiences with fast-and-frugal decision trees indicate the non-smooth transition around $\tau=25$ is due to their structural limitations.
>
> >Could you better clarify and contrast how the empirical performance differs between these two specific data-generating processes [observational vs. randomized trial data]?
>
> There are at least two main dimensions to this contrast: one is sample size and the other is the nature of partial identifiability.
>
> First, observational studies typically provide a lot more samples than a randomized trial. On the face of it, this would favor observational studies. Second, however, the role that unmeasured factors $U$ play in confounding versus selection are fundamentally different (illustrated in Fig. 2). For fixed samples size, therefore, it is not possible to draw meaningful conclusions about how the empirical performance differs because the effects of unmeasured confounding or selection factors can be made arbitrarily large in any example.
>
> That said, by comparing the derived weights in (10) versus (11) it can be seen that it is easy to make the method conservative in the case of unmeasured selection by increasing $\Gamma_s$ in (7).
>
> >This severely limits the paradigm's versatility, making it exceedingly difficult to scale or generalize to the more expressive, high-capacity model architectures (e.g., deep neural networks) that are standard in modern policy learning
>
> In this context, with high-stakes decisions, clinical experts prefer interpretable policy classes $\Pi$, such as decision trees, over the uninterpretable but expressive alternatives. (See also the discussion in Rudin (2019) cited on p.2.)
>
> >It is debatable whether this strict control consistently outperforms models that simply ignore unmeasured confounding $U$
>
> Models that simply ignore $U$ cannot certify any treatment risk control and would therefore not be commensurable with models that do so.
>
> >benchmark against standard Empirical Risk Minimization (ERM) approaches that optimize for population risk without the treatment risk constraint, in order to explicitly quantify the cost of this constraint.
>
> This benchmark is actually provided in Figure 4, where the constraint for ERM becomes inactive. This results in more than a quarter of treatments having no benefit. We believe it is for clinical experts and decision-makers to reason about appropriate trade-offs in the use of health care resources. That is what our proposed method offers to them.

---

> > ### Author Rebuttal · Reviewer_XRtA · 2026-04-02
> >
> > Thank you! I have no further comments.

---

### Official Review · Reviewer_aJD5 · 2026-03-11

**Soundness:** 4
**Presentation:** 4
**Significance:** 3
**Originality:** 3
**Overall Recommendation:** 5
**Confidence:** 2

**Summary:**

- Authors propose a method to treat ‘treatment risk’ in optimal treatment policy calculation using partial identification
- They motivated and formulate the problem well and contextualise it sufficiently in established literature
- Their proposed method performs well in benchmarks

**Compliance With Llm Reviewing Policy:**

Affirmed.

**Key Questions For Authors:**

- Notation: “L” is AFIAK know mostly used for covariates, and Y for outcomes, maybe change to Y for outcomes?
- Is your example in the question page top right contrived or not? Would different setups yield different results?
- What is your definition of undirected edges in Figure 2, e.g. between U and X?

**Limitations:**

- on page 1, right column line 42, there might be value in emphasising just how hard/impossible it is to 'assess' unmeasured confounding, e.g. which in general cannot be tested for/empirically validated or something similar

**Strengths And Weaknesses:**

Strengths
- Formulation seems clear, though I am not an expert
- the synthetic data experiments are introduced well, contrasting both cases with no confounding and unmeasured confounding
- The STAR trial showcases the method well

Weaknesses
- I lack expertise to spot obvious weaknesses, though am able to place this well in the causal inference literature

---

> ### Author Rebuttal · Authors · 2026-03-29
>
> We thank the reviewer for the feedback and questions.
>
> >$L$ is AFIAK know mostly used for covariates, and Y for outcomes, maybe change to Y for outcomes?
>
> The notational style does vary a bit across literatures. $X$ for covariates and $A$ for action is a common choice in the cited literature on contextual bandits. It is also common to use a negative loss or reward $R$ as an outcome variable. Here we have chosen loss $L$ which fits the more widespread risk minimization paradigm and more over is appropriate in high-stakes decision making.
>
> >Is your example in the question page top right contrived or not? Would different setups yield different results?
>
> While the example given at the beginning of the paper is stylized, the trade-off between treatment risk $T(\pi)$ and population risk $R(\pi)$ that it illustrates is near-universal. (This is hinted at by the decomposition of $R(\pi)$ in Eq. (2).)
>
> In other words, the trade-offs illustrated by the two curves also appear under different setups but their sharpness will vary. (This is empirically illustrated by the contrast between the STAR and IST data shown in Figs. 5 and 11.)
>
> >What is your definition of undirected edges in Figure 2, e.g. between U and X?
>
> It means that the causal path between $U$ and $X$ is unknown and remains unspecified. This means (1), (4) and (5) leave $p(X,U|S=s)$ as unfactorized since the causal direction is unknown.

---

> > ### Author Rebuttal · Reviewer_aJD5 · 2026-04-02
> >
> > I see, apologies, I wasn't aware that L is commonly used in bandits, though I can see it now as "loss".
> >
> > A definition of which undirected edge framework you are using might be useful for the reader.
> >
> > I have no further concerns.

---

### Official Review · Reviewer_qgoA · 2026-03-13

**Soundness:** 3
**Presentation:** 3
**Significance:** 4
**Originality:** 3
**Overall Recommendation:** 5
**Confidence:** 3

**Summary:**

The manuscript studies how to learn treatment-allocation policies that reduce population risk while explicitly controlling the fraction of treated individuals who receive no benefit, which the paper terms as treatment risk. This target is clinically meaningful in settings where overtreatment is costly or harmful. The proposed method learns a nominal policy under a user-specified miscalibration bound \Gamma, then uses sample splitting and an upper confidence bound to certify that treatment risk stays below a tolerance \tau with probability at least 1−\alpha, even when the risk is only partially identifiable. The framework covers both observational studies with unmeasured confounding and randomized trials with selection or transportability bias.

**Compliance With Llm Reviewing Policy:**

Affirmed.

**Key Questions For Authors:**

How often does the equality condition in Theorem 4.3 hold in practice for the fast-and-frugal tree search, and how sensitive is certification when it only holds approximately?

**Limitations:**

yes

**Strengths And Weaknesses:**

Strengths:
This paper is well-motivated, contributing to overtreatment-aware or confounding-robust policy learning under partial identification. Interestingly, it unifies two important sources of partial identification, including hidden confounding in observational studies and selection bias in randomized trials within the same odds-miscalibration framework. The proposed method is well aligned with the stated goal, and finite-sample guarantee seems good.

Weaknesses:
The paper claims that "problems of overtreatment and low-value treatment allocations have remained unexplored in the literature on policy learning,” (L208) which could be over-stated, as its related-work section cites prior work on constrained policy learning, partial identification, and counterfactual harm. The assumption of the main theorem that the nominal policy must achieve the training constraint with equality may be fragile in practice.

---

> ### Author Rebuttal · Authors · 2026-03-29
>
> We thank the reviewer for the positive feedback.
>
> >How often does the equality condition in Theorem 4.3 hold in practice for the fast-and-frugal tree search
>
> The equality-condition held in all the various examples and depths of fast-and-frugal trees we have tried. Thus it has not been restrictive and is moreover an easy check to perform in practice.
>
> >how sensitive is certification when it only holds approximately?
>
> We have not constructed cases when the condition fails, but given the conservative nature of the proof, we would suspect a graceful degradation.

---

> > ### Author Rebuttal · Reviewer_qgoA · 2026-04-05
> >
> > I found the response is still fairly anecdotal. For my first question, it would be helpful if the authors could discuss the limitations in the last section. For my second question, the question of how certification behaves when the condition only approximately holds is still unresolved, as the rebuttal provides only conjecture.

---

> > > ### Author Response · Authors · 2026-04-07
> > >
> > > **Reg. Discussion about equality constraint**
> > >
> > > We are in agreement: with an added page, it would be good to discuss the possible limitations raised by the reviewer. Specifically, the result in Theorem 4.3 holds when $\pi(X; \tau)$ in (12) achieves the empirical constraint with equality. While this condition is readily verifiable, there are indeed cases when only strict inequality of the empirical constraint can be achieved. This will depend on the nature of the data-distribution $p$, the weight model $\hat{p}(A|X)$ or $\hat{p}(S|X)$ and the capacity of the chosen policy class $\Pi$.
> > >
> > > After the reviewer’s prompting, our conclusions from using fast-and-frugal decision trees (FFT) could be revised and summarized as follows: The equality condition holds throughout the synthetic examples. But in the case with empirical data (Figs. 5 and 11), we see that when $\tau$ becomes sufficiently low the learned FFT policy reverts into a `treat-none’ policy while equality of the empirical constraint is no longer possible. In these cases the treatment risk is controlled according to (8) but by a conservative policy. This is, of course, fully transparent to the user.
> > >
> > > In the case when only strict inequality holds but the learned policy is not `treat-none', no certification is possible with the methodology in the paper. This is a limitation but the method also indicates the case to the user.
> > >
> > > We will include these relevant points raised by the reviewer in Section 6 of the revised paper.

---

### Official Review · Reviewer_SV6i · 2026-03-13

**Soundness:** 3
**Presentation:** 3
**Significance:** 2
**Originality:** 2
**Overall Recommendation:** 2
**Confidence:** 5

**Summary:**

This paper studies treatment allocation under partial identifiability, with the goal of minimizing overall population risk while controlling the proportion of treated individuals who do not benefit from treatment. The paper considers both observational studies with unmeasured confounding and randomized trials with selection bias between trial and target populations. Under a bounded miscalibration model, the authors derive upper bounds on both population risk and treatment risk, then use sample splitting and an upper confidence bound to learn a policy that controls treatment risk with high probability in finite samples. The method is illustrated using synthetic experiments and the STAR dataset.

**Compliance With Llm Reviewing Policy:**

Affirmed.

**Final Justification:**

My final recommendation is Reject. The paper is clearly written and technically reasonable overall. However, I remain less convinced by the originality and significance. The core contribution still feels closer to a careful combination of partial-identification bounds, sensitivity analysis, and finite-sample calibration than to a major methodological advance, and the empirical evidence remains more illustrative than confirmatory. In particular, the main certification guarantee is only meaningful relative to a user-specified $\Gamma$, and the real-data evaluations do not fully validate the most important quantity of interest. The rebuttal was helpful in clarifying the practical interpretation of $\Gamma$ and the distinction between treatment risk and counterfactual harm, but it did not fully resolve my main concerns about external validation and empirical positioning. Overall, while I think the paper is thoughtful and competently executed, I remain below the acceptance bar.

**Key Questions For Authors:**

1. The main certification guarantee is conditional on the miscalibration level. Can the authors clarify more explicitly how they expect practitioners to choose Γ in realistic applications, and how sensitive the learned policies are to misspecification of this parameter?

2. The real-data experiment is presented as an illustration, but the key treatment-risk quantity is not directly verifiable there. Do the authors have a stronger real-world validation strategy, or additional evidence that the learned policies are useful beyond the synthetic setting?

3. How does the proposed method compare empirically with existing constrained or safe policy learning methods under partial identification or sensitivity analysis? Stronger baselines would help clarify the practical value of the proposed approach.

4. The paper emphasizes treatment risk as the proportion of treated individuals without benefit. Can the authors better clarify how this notion relates to alternative notions such as counterfactual harm or no-harm criteria, and when one should prefer this definition?

**Limitations:**

The paper discusses some limitations, but I think it should be more explicit that the certification guarantee is only valid relative to a user-specified bound Γ on unmeasured confounding or selection bias. Since Γ is not identifiable from the observed data, the guarantee is necessarily conditional.

**Strengths And Weaknesses:**

Soundness: The setup is clear, the upper-bound formulation under bounded miscalibration is reasonable, and the sample-splitting argument for finite-sample certification is coherent. My main reservation is that the guarantee is only conditional on the specified miscalibration level Γ being appropriate. Since Γ is fundamentally unverifiable in the settings of interest, the practical meaning of the certification is somewhat weaker than the paper’s framing may initially suggest. In addition, the real-data experiment is more illustrative than confirmatory, since the key treatment-risk quantity cannot truly be validated there.

Presentation: The paper is generally well written and easy to follow.

Significance: I can see the motivation for the problem, especially in settings where overtreatment is itself a meaningful concern. The paper also has a practically sensible emphasis on interpretable policies. However, I was not fully convinced of the significance of the contribution. The main technical idea feels closer to a careful combination of partial-identification bounds, sensitivity analysis, and finite-sample calibration than to a major methodological advance, and the empirical evidence is not yet strong enough to suggest broad impact beyond this fairly specific setting.

Originality: The treatment-risk perspective is a useful framing, and bringing finite-sample certification into this partially identified policy-learning setting is a reasonable contribution. That said, the overall novelty feels moderate rather than strong. Much of the paper builds on existing ideas from sensitivity analysis, constrained policy learning, and risk-controlling prediction, and the resulting method feels more incremental than fundamentally new.

---

> ### Author Rebuttal · Authors · 2026-03-29
>
> We thank the reviewer for the constructive feedback, and provide our replies below.
>
> >Can the authors clarify more explicitly how they expect practitioners to choose $\Gamma$ in realistic applications, and how sensitive the learned policies are to misspecification of this parameter?
>
> Indeed, as stated following eq. (8), the certification guarantee is conditionally valid for any degree of odds miscalibration *up to* $\Gamma$. (This is also repeated in Theorem 4.3, in the experiment in Section 5.1, as well as in the concluding discussion in Section 6.)
>
> The choice of credible values of $\Gamma$ is raised after eq. (7) (paragraph starting on p.3, 1st col, rows 162 to 113 on the following page): Following the discussions in the cited works, we demonstrate two means of choosing $\Gamma$ in Appendix A.2. The first method assesses the model misspecification of nominal odds (lines 579-586), which is illustrated in Figure 7.a. The second method assesses the possible impact of unmeasured confounders by considering an omitted covariate $X_k$ as $U$, illustrated in Figure 7.b. Domain knowledge guides which omitted covariate would serve as credible benchmarks. In both cases, credible choices for $\Gamma$ are obtained, either in relation to misspecification and potential unmeasured confounding. In our interaction with clinical experts and epidemiologists both methods were found to be transparent means of drawing robust conclusions from data.
>
> $\Gamma$ increases the tightness of constraint (12) via (10) or (11), so that learned policies become increasingly conservative with the degree of miscalibration $\Gamma$. This remains the case whether or not $\Gamma \geq \Gamma_0$ (valid) or $\Gamma < \Gamma_0$ (misspecified) for some true unknown $\Gamma_0$. This is illustrated in Section 5.1, where we show policies in Figure 3 under $\Gamma=1$ and then refer to Appendix A.3 (Figure 9) for a contrast with $\Gamma=2$. We see that when the degree of miscalibrated odds increases to a factor 2, the recommended age for receiving $A=1$ is lowered uniformly across $\tau$.
>
> >Do the authors have a stronger real-world validation strategy, or additional evidence that the learned policies are useful beyond the synthetic setting
>
> While certified real-world *validation* requires expensive trials, the conclusion from our discussions with clinical experts is encouraging: The utility is deemed high because the learned policies have statistically certifiable risks *and* are interpretable. For the clinical expert, the combination is what renders the policy *trustworthy* and possible to compare with existing clinical guidelines. Thus the learned policies could constitute a first step towards actually improving guidelines for existing treatment allocations.
>
> >How does the proposed method compare empirically with existing constrained or safe policy learning methods under partial identification or sensitivity analysis?
>
> To the best of our knowledge, most work on policy learning under partial identifiability or constraints, which is broadly represented in Section 3, is focused on continuous (and not binary) health outcomes. Thus they are not amenable to empirical comparison with our method. Moreover, none of the methods offer any finite sample valid control of some risk. We will highlight these differences in the revised manuscript.
>
> >The paper emphasizes treatment risk as the proportion of treated individuals without benefit. Can the authors better clarify how this notion relates to alternative notions such as counterfactual harm or no-harm criteria, and when one should prefer this definition?
>
> We can now elaborate on this, since there was not quite enough space in the submitted manuscript.
>
> First, recall that the `treatment risk’ or proportion of treated individuals without benefit is $T(\pi) = \mathbb{P}_{\pi}(L=1|A=1)$.
>
> In contrast, the counterfactual notion of harm starts with the idea that a patient, with covariates $X$, always has *two* parallel losses $L(0)$ and $L(1)$ corresponding to receiving $A=0$ and $A=1$, respectively. Obviously, only *one* of them can be observed. Being *negatively affected* by a treatment $A=1$ is then defined as: $L(1)=1$ while *simultaneously* $L(0)=0$. The counterfactual fraction of negatively affected by a policy is defined as $FNA(\pi)$ = $\mathbb{E}$$[\mathbb{P}(L(0)=0, $L(1)=1|X)\pi(X)]$.
>
> Since only one of the losses, $L(0)$ *or* $L(1)$ can be observed, $FNA(\pi)$ is unobservable and can only be *upper bounded*. Thus $FNA(\pi)$ is philosophically interesting and relevant, but *cannot* be validated even in principle in a clinically relevant setting. In contrast, the treatment risk $T(\pi)$ is easily communicated as the fraction of treatments with no benefits, it can be controlled, and the control *can* be validated. It is therefore of value in clinical practice and can help guide decision-makers to strike trade-offs in the use of finite resources to improve health outcomes.

---

> > ### Author Rebuttal · Reviewer_SV6i · 2026-04-03
> >
> > Thank you for the clarification. The rebuttal helps, especially regarding the practical interpretation of \Gamma and the distinction between treatment risk and counterfactual harm. However, I still have some residual concerns about external validation and empirical positioning. In particular, I am still not fully convinced that the paper provides sufficiently strong real-world evidence, or a strong enough empirical comparison to closely related constrained/safe policy learning approaches, to justify the broader practical claims.

---

> > > ### Author Response · Authors · 2026-04-04
> > >
> > > **Reg. external validation and real-world evidence**
> > >
> > > For any off-policy learning method a complete external validation would require setting up an experimental trial, which would not just be costly but require ethical approvals. We are certain the reviewer agrees that this is beyond the scope of an ICML paper on off-policy learning.
> > >
> > > The standard empirical validation in the off-policy learning literature is therefore to use synthetically generated scenarios based on real-world data. This is what we have followed, using the STAR and International Stroke Trial datasets in Sections 5.2 and A.2, respectively. Such empirical studies are indeed meant to be illustrative rather than conclusive, motivating the need for further experimental studies. Section 3 provides references for off-policy learning methods that are evaluated accordingly.
> > >
> > > If the reviewer has suggestions for additional open datasets that could be used to strengthen the validation of the methodology, we would gladly consider them in future work.
> > >
> > > **Reg. empirical comparisons to constrained/safe policy learning approaches**
> > >
> > > We have provided two clear baselines available in the case of binary outcomes: The first is empirical risk minimization ERM $\min_{\pi} \hat{\mathbb{E}}_{\pi}[L]$, which is the dominant paradigm in machine learning. The second is ERM with a constraint on the empirical treatment risk. The latter reduces to the former as the risk tolerance $\tau$ approaches 1. Both baselines are provided in the discussion around Figures 4 and 8.
> > >
> > > For clarity, let’s take the constrained and/or safe policy learning approaches cited in Section 3 in turn:
> > >
> > > 1. Kitagawa & Tetenov (2018) and Athey & Wager (2021) consider only structural constraints on the policy class $\Pi$ such as a treatment budget. They assume full identifiability in a case with a single primary outcomes in $\mathbb{R}$.
> > >
> > > 2. Wang et al. (2018) and Doubleday et al. (2022) assume full identifiability in a case with two *different* outcomes, both in $\mathbb{R}$, that they seek to balance.
> > >
> > > 3. Kallus (2022) and Li et al. (2023) seek to asymptotically constrain an unidentifiable harm measure based on two counterfactual outcomes, both in {$0, 1$}.
> > >
> > > In other words, the outcomes of interest *and* the constraints considered above are not jointly commensurable with the problem tackled in our paper. Moreover, none of the methodologies above achieve finite-sample valid control over the constraints they seek to satisfy. If the reviewer has suggestions for other comparable approaches, we would also consider them.

---

### Decision · Program_Chairs · 2026-04-30

**Decision:**

Accept (regular)

**Comment:**

This paper studies treatment allocation under partial identifiability, aiming to minimize overall population risk while controlling the proportion of treated individuals who do not benefit from the treatment. Based on the reviewers’ comments and the authors’ detailed rebuttal, we are inclined to recommend acceptance of the paper.

Nevertheless, the reviewers’ feedback and our own careful reading highlight several notable weaknesses that should be addressed in the final version:
* Unclear problem setting. It remains ambiguous whether $L$ is observed. The role of the notation $S$ is also not well motivated—specifically, whether it is required for both experimental and observational data. In addition, the identification assumptions are not stated clearly. These foundational aspects should be clarified.
* Lack of clarity in key concepts. The distinction between the “treatment risk” considered in this paper and related notions in the literature—such as counterfactual harm, fraction negatively affected, or individual risk [1,2,3]—is not sufficiently articulated. A more detailed comparison and discussion would strengthen the positioning of the work.
* Incomplete presentation of main contributions. The paper does not clearly enumerate its primary contributions. As currently written, the contributions appear limited, which undermines the perceived novelty and impact of the work. A clearer and more explicit statement of contributions is needed.

We encourage the authors to carefully incorporate all other reviewer's suggestion into the final revision.

[1] What’s the harm? sharp bounds on the fraction negatively affected by treatment. NIPS, 2022

[2] Quantifying Individual Risk for Binary Outcomes, arXiv, 2024

[3] Safe Individualized Treatment Rules with Controllable Harm Rates, arXiv, 2025